# Functional Characterization of Aldehyde Dehydrogenase in *Fusarium graminearum*

**DOI:** 10.3390/microorganisms11122875

**Published:** 2023-11-28

**Authors:** Lei Tang, Huanchen Zhai, Shuaibing Zhang, Yangyong Lv, Yanqing Li, Shan Wei, Pingan Ma, Shanshan Wei, Yuansen Hu, Jingping Cai

**Affiliations:** College of Biological Engineering, Henan University of Technology, Zhengzhou 450001, China; tangleixs@163.com (L.T.); shbzhang@163.com (S.Z.); lvyangyong2011@163.com (Y.L.); liyq4224@126.com (Y.L.); weishansd2014@163.com (S.W.); mapingan@haut.edu.cn (P.M.); shanshanwei1211@126.com (S.W.); hys308@126.com (Y.H.); caijp163@163.com (J.C.)

**Keywords:** enzymes, *Fusarium*, mutants, abiotic stress, deoxynivalenol, pathogenicity, gene expression

## Abstract

Aldehyde dehydrogenase (ALDH), a common oxidoreductase in organisms, is an aldehyde scavenger involved in various metabolic processes. However, its function in different pathogenic fungi remains unknown. *Fusarium graminearum* causes Fusarium head blight in cereals, which reduces grain yield and quality and is an important global food security problem. To elucidate the pathogenic mechanism of *F. graminearum*, seven genes encoding ALDH were knocked out and then studied for their function. Single deletions of seven ALDH genes caused a decrease in spore production and weakened the pathogenicity. Furthermore, these deletions altered susceptibility to various abiotic stresses. *FGSG_04194* is associated with a number of functions, including mycelial growth and development, stress sensitivity, pathogenicity, toxin production, and energy metabolism. *FGSG_00139* and *FGSG_11482* are involved in sporulation, pathogenicity, and SDH activity, while the other five genes are multifunctional. Notably, we found that *FGSG_04194* has an inhibitory impact on ALDH activity, whereas *FGSG_00979* has a positive impact. RNA sequencing and subcellular location analysis revealed that *FGSG_04194* is responsible for biological process regulation, including glucose and lipid metabolism. Our results suggest that ALDH contributes to growth, stress responses, pathogenicity, deoxynivalenol synthesis, and mitochondrial energy metabolism in *F. graminearum*. Finally, ALDH presents a potential target and theoretical basis for fungicide development.

## 1. Introduction

Aldehyde dehydrogenase (ALDH), a common oxidoreductase, is an aldehyde scavenger that catalyzes the irreversible oxidation of various aromatic and aliphatic aldehydes into their corresponding carboxylic acids. The ALDH Gene Nomenclature Committee has established criteria for classifying ALDH into 24 families across all taxa [1]. It is widely found in both prokaryotes and eukaryotes. In bacteria, *Bacillus licheniformis* ALDH plays a key role as a biocatalyst with significant resistance to water-soluble organic solvents [2]. In plant species, 14 different ALDH families have been identified [3]. Genomic analyses of the ALDH supergene family has identified 20 ALDH genes in rice [4], 14 in *Arabidopsis* [5], and 23 in grapes [6]. Nineteen ALDHs have been identified in humans and have been divided into 11 families [7,8]. To date, six ALDHs have been identified and annotated in the *Saccharomyces* Genome Database [9]. Sixteen ALDHs have also been identified in the filamentous fungus *Magnaporthe oryzae* and divided into nine branches [10].

ALDH plays important roles in bacteria, plants, mammals, and fungi. The NADP (+)-dependent ALDH present in *Zymomonas mobilis* converts hazardous acetaldehyde into acetic acid, thus providing a detoxifying effect [11]. Plant ALDHs have various functions, with multiple roles having been identified in rice. For example, OsALDH7 maintains seed viability [12], and OsALDH2a is involved in submergence tolerance [13]. OsALDH2b is necessary for male reproductive development and inhibits tapetal programmed cell death [14]. Grape ALDH genes are essential for grape development, and some have been observed to be reactive to salt stress or drought [6]. Comparative genomic analyses of the ALDH gene superfamily in *Arabidopsis thaliana* have demonstrated that ALDH is a critical factor in hypoxia tolerance [15]. The role of ALDH in both human and yeast cells has also been extensively studied, as ALDH removes detrimental aldehydes. There are various members of the ALDH family in humans. ALDH2 is a vital component of the ALDH family in humans and is essential for the elimination of acetaldehydes and other aldehydes. ALDH2 has been identified as a key factor in the neuroprotective system and in the prevention and management of cardiovascular diseases [16,17]. Several studies have purified and characterized the relevant yeast ALDH in *Saccharomyces cerevisiae*. ALDH has a major effect on acetaldehyde metabolism in yeast [18]. ALDHs are indispensable in yeast cells, as they confer tolerance to various stressors, including high ethanol concentrations, osmotic stress, and oxidative stress [18,19,20].

The function of ALDH in filamentous ascomycetous fungi remains largely unknown. The deletion of methylmalonate semialdehyde dehydrogenase (*MoMSDH*) has been found to have a drastic effect on conidiation, appressoria formation, and pathogenesis in *Magnaporthe oryzae* [10]. Previous studies found that the inactivation of MoP5CDH and MoKDCDH had a major impact on the pathogenesis of *M. oryzae* [21]. Additionally, knockout of the ALDH gene in *Fusarium oxysporum* has been suggested to improve ethanol yield [22].

*Fusarium graminearum* is the cause of Fusarium head blight (FHB), a fungal disease of cereal grains that has been wreaking havoc across the globe. This disease reduces cereal yield and also leads to grain contamination with mycotoxins that are dangerous to human and animal health [23]. To successfully control FHB, it is very important to study the mechanisms that regulate the growth of the pathogen and the synthesis of its mycotoxins. With the completion of the *F. graminearum* genome sequencing, many gene functions and regulatory mechanisms have been identified. Deoxynivalenol (DON) is the main mycotoxin produced by *F. graminearum*, which plays a role as a pathogenicity factor during fungal infection of plants [24]. DON synthesis is affected by many environmental conditions, such as the availability of carbon and nitrogen sources, H_2_O_2_, and salt stress [25]. Using high-throughput techniques to analyze the transcriptome of *F. graminearum* under salt stress, we found that salt stress regulates the expression of several ALDHs. However, the relationship between ALDH and stress, regulation of DON synthesis, and the pathogenicity of *F. graminearum* remain unclear.

In this study, nine annotated genes coding for aldehyde dehydrogenase (*FGSG_04194*, *FGSG_00139*, *FGSG_11482*, *FGSG_05375*, *FGSG_00979*, *FGSG_02160*, *FGSG_05831*, *FGSG_02273*, and *FGSG_01759*) were identified in *F. graminearum*. It is unclear why efforts to knockout *FGSG_ 02273* and *FGSG_01759* did not succeed. An evaluation of the biological functions of the seven ALDHs indicated that they play crucial roles in the conidial production, DON regulation, stress responses, and pathogenicity of *F. graminearum*. This study provides a theoretical basis to deepen our knowledge of the pathogenesis of *F. graminearum*, improve our understanding of the function of fungal ALDH protein family genes, and identify new targets for the development of fungicides.

## 2. Materials and Methods

### 2.1. Strains and Culture Conditions

The *F. graminearum* wild-type PH-1 parental strain used to construct all the mutant strains was kindly donated by Prof. Zonghua Wang, and seven ALDH-deletion mutants (Δ*Fg04194*, Δ*Fg00139*, Δ*Fg11482*, Δ*Fg05375*, Δ*Fg00979*, Δ*Fg02160,* and Δ*Fg05831*) and a complementation strain of *FGSG_04194* (Δ*Fg04194-C*) were constructed and preserved in this study. All strains were preserved as conidial suspensions in a CMC medium that was composed of 15 g/L sodium carboxymethylcellulose, 1 g/L NH_4_NO_3_, 1 g/L KH_2_PO_4_, 0.5 g/L MgSO_4_·7H_2_O, 1 g/L yeast extract, and 20% glycerol, and stored at −80 °C. The strain was activated by culturing it on a solid complete medium (CM) composed of 6 g/L yeast extract, 6 g/L casamino acid, 10 g/L sucrose, and 20 g/L agar at 28 °C for 72 h.

### 2.2. Identification of the ALDH in F. graminearum

The *Saccharomyces cerevisiae* ALD2 (NP_013893.1) protein sequence was used as a target protein, and the annotated genes coding for aldehyde dehydrogenase in *F. graminearum* were identified through BLAST in the NCBI database (https://www.ncbi.nlm.nih.gov/gene, accessed on 6 October 2023). The ALDH homologous protein of *F. graminearum* was analyzed by multiple alignment using MEGA-X software, and a phylogenetic tree was constructed by the neighbor-joining method with a bootstrap of 1000. Analysis of the conserved domains of ALDH proteins was conducted through the NCBI Conserved Domains database (https://www.ncbi.nlm.nih.gov/Structure/cdd/wrpsb.cgi, accessed on 7 October 2023).

### 2.3. Construction of Deletion and Complementary Mutants

Homologous recombination was employed to knock out each target gene. The target gene’s upstream fragment and downstream fragment were amplified using specific primers 1F/2R and 3F/4R, respectively. By using the pCX62 plasmid as the template, the left fragment and the right fragment of the hygromycin resistance gene were amplified with HYG/F-HY/R and YG/F-HYG/R primers, respectively. The split-marker technique was employed to create a gene replacement construct for the ALDH gene in accordance with the protocol described in [26]. Following the protocol described by Hou et al. [27], *F. graminearum* protoplast preparation and fungal transformation was carried out. The transformants were identified through PCR with primers 5F/6R, 7F/H855R, and H856F/8R. The positive transformants was further confirmed by real-time PCR with primer QF/QR. For the complementation, FGSG_04194-GFP was generated through the amplification of a 2614 bp fragment, which included the 1007 bp *FGSG_04194* coding sequence and a 1607 bp promoter region, using primer C04194F/R. The PCR product of 2614 bp was inserted into the pKNTG vector, which included the GFP allele and the neomycin gene as a selection marker. The constructs that were confirmed by sequencing were successfully transformed into the Δ*Fg04194* mutant. To verify the complementary transformants, a PCR screening was conducted with C04194F/R and GFP-Hind3-F/GFP-BamHI-R primers. All of the primer sequences are listed in Appendix A.

### 2.4. Fungal Growth and Conidia Production Analysis

To observe the cultural characters of strains, the *F. graminearum* PH-1 strain and ALDH-deletion mutants were cultivated on CM solid media at 28 °C for 72 h. The diameter of each colony was measured in two perpendicular directions [28]. For conidia production, four discs of 5 mm diameter, taken from the margin of a 3-day-old colony of each strain, were cultured in CMC liquid media at 28 °C and 150 rpm for 72 h. The conidia were then counted using a hemocytometer and observed under a fluorescence microscope (Bio-Tek Epoch, Beijing, China) after being stained with 10 μg/mL Calcofluor White (CFW). The conidia suspension was then collected, after which the conidia concentration was adjusted to 10^6^ cfu/mL. The conidia were then incubated at 28 °C and 80 rpm for 24 h. To assess the germination rate of conidia, a Nikon light microscope (Nikon, Tokyo, Japan) was employed to observe a minimum of 100 conidia per strain at 0, 2, 4, 6, 8, and 24 h. For each strain, three biological replicates were employed.

### 2.5. Response of the Knockdown Mutants to Stress

To analyze the susceptibility of ALDH genes to various abiotic stressors, including oxidative stress (H_2_O_2_ and acetaldehyde), osmotic stress (sorbitol and NaCl), the cell-membrane-destructing factor (SDS), fungicides (alcohol and citral), and the cell-wall-destructing factor (CR) in *F. graminearum*, fresh mycelial blocks of PH-1 and ALDH-deletion mutants with the same diameter were attached to the center of CM solid plates containing 0.05% H_2_O_2_, 0.1% acetaldehyde, 3% sorbitol, 0.01% sodium dodecyl sulfate (SDS), 1 M NaCl, 800 μg/mL Congo Red (CR), 2.5% alcohol, and 0.02% citral and incubated at 28 °C for 72 h. CM medium without stressors was used as the control. The diameters of the colony were recorded and documented through photographs. Three replicates were performed for all of the stress experiments.

### 2.6. Pathogenicity and DON Synthesis Analysis

Zhengmai 366, a wheat cultivar that is moderately susceptible to *F. graminearum* infection developed by the Wheat Research Institute of the Henan Academy of Agricultural Sciences, was chosen for analysis of the pathogenicity and production of DON by this fungus. For pathogenicity analysis, conidia from 3-day-old CMC cultures were harvested and resuspended in sterile distilled water to a concentration of 10^6^ cfu/mL. A total of 10 μL of conidium suspensions were injected into the flowering wheat heads at the center of the wheat spikelets, and wheat spikelets that were injected with sterile water were used as a control. For each strain, 10 replicates were introduced into the spikelets for examination. The infected spikelet was examined 10 days after inoculation [25]. For DON production analysis, four discs of 5 mm diameter, taken from the margin of a 3-day-old colony of the wild-type strain and mutant strains [29], were inoculated into 100 g of sterilized wheat grains and incubated at 28 °C with regular shaking for 21 days. The deoxynivalenol ELISA rapid test kit (Huaan Magnech, Beijing, China) was used to determine the DON content. The samples were handled and assay procedures were completed in accordance with the manufacturer’s instructions. Three replicates were performed for each strain.

### 2.7. Determination of ALDH Activity, SDH Activity, and ATP Content

The wild-type strain and mutants were cultured in CM liquid media at 28 °C and 150 rpm for 72 h. Mycelia were then collected and rinsed with PBS (0.1 M, pH 7.4) to remove impurities. A total of 0.1 g of mycelium was measured after being blotted dry with sterile filter paper, and 1 mL of the extract was added to the ice bath homogenization and then centrifuged at 10,000× *g* and 4 °C for 20 min, and the supernatant was placed on ice to be measured for the activity of ALDH and succinate dehydrogenase (SDH). For the adenosine triphosphate (ATP) content test, the supernatant was transferred to another 1.5 mL centrifuge tube, mixed with 500 μL of chloroform, and centrifuged at 4 °C for 3 min. Afterward, the supernatant was collected and stored on ice for testing. The treated samples were assayed for acetaldehyde dehydrogenase activity, SDH activity, and ATP content using an assay kit (Solarbio, Beijing, China), following the manufacturer’s instructions. For each strain, three separate biological replicates were assessed.

### 2.8. Transcriptomic Analysis

The wild-type strain and Δ*Fg04194* mutant were inoculated in CM liquid media and incubated at 28 °C and 150 rpm for 72 h. Mycelia were then collected and sent for transcriptome sequencing, with three replicates per strain. Following RNA extraction, purification, and library construction, paired-end sequencing was conducted using a next-generation sequencing platform based on Illumina technology. A differential analysis of gene expression was performed using DESeq, with an expression difference multiplicity of |log2FoldChange| > 1 and *p* < 0.05 indicating statistically significant results [30]. Gene Ontology (GO) and Kyoto Encyclopedia of Genes and Genomes (KEGG) enrichment analyses were carried out by utilizing topGO and cluster Profiler. The raw RNA sequencing (RNA-seq) data obtained in this study were uploaded to the NCBI Sequence Read Archive (BioProject ID: PRJNA983120). The obtained data were analyzed to identify differentially expressed genes (DEGs) related to DON biosynthesis and *F. graminearum* pathogenicity. The expression of some of the DEGs was verified using real-time quantitative PCR.

### 2.9. Real-Time PCR Validation

Fresh mycelia of the wild-type strain and deletion mutants were inoculated in CM liquid media and incubated at 28 °C and 150 rpm for 72 h. Mycelia were then collected, total RNA was extracted, and the reverse transcription PrimeScript TM II 1st Strand cDNA Synthesis Kit (TaKaRa, Beijing, China) was used in accordance with the manufacturer’s stipulations. The cDNA was synthesized, verified by amplifying the internal reference gene tubulin, and stored at −20 °C. The expression of *F. graminearum* genes was assessed through a fluorescent qPCR, using the ChamQ Universal SYBR qPCR Master Mix Kit (Vazyme, Nanjing, China), and the CT values of the internal reference gene, tubulin, and target gene were obtained. The relative expression of the target genes was determined by the 2^−ΔΔCt^ method, and the primers are listed in Appendix A.

### 2.10. Analysis of Subcellular Localization and Lipid Droplet Accumulation

To ascertain the subcellular localization of FGSG_04194 and the changes in the accumulation of lipid droplets (LDs) in the Δ*Fg04194* strain, we stained PH-1, Δ*Fg04194*, and Δ*Fg04194-C* conidia and hyphae incubated at 28 °C for 72 h with 10 μg/mL Nile Red. Green fluorescent protein (GFP) and Nile Red fluorescence signals of the conidia and hyphae were observed using a laser scanning confocal microscope (Nikon, Tokyo, Japan).

### 2.11. Statistical Analysis

Graphs were drawn using GraphPad Prism 8.0.1 software. Statistical analysis was conducted using IBM SPSS Statistics 20 software. A one-way analysis of variance and Fisher’s least significant difference test were applied to assess the significance.

## 3. Results

### 3.1. Analysis of ALDH Homologous Sequences in F. graminearum

Utilizing the BLASTP on the *F. graminearum* PH-1 protein database with the *S. cerevisiae* ALD2 (NP_013893.1) protein as a query, 29 homologous proteins were discovered. Phylogenetic analysis reveals that nine proteins annotated as aldehyde dehydrogenase are clustered in the same branch, three proteins annotated as succinate semialdehyde dehydrogenase (FGSG_11843, FGSG_06752, and FGSG_04196) are grouped together, and two proteins annotated as methylmalonate semialdehyde dehydrogenase (FGSG_00490 and FGSG_01826) are clustered in the same branch, while the other hypothetical proteins are located in separate branches (Figure 1A). It can be inferred that the homologous proteins in *F. graminearum* have evolved to acquire different functions, despite their shared common functions. An analysis of the conserved domain of the nine proteins annotated as aldehyde dehydrogenase was conducted (Figure 1B). Most of the ALDHs were found to have relatively conserved ALDH-SF structural domains and belonged to the NAD(P)+ superfamily of ALDHs. This suggests that ALDHs play a significant role in detoxification. The conserved domain, ALD2, of the FGSG_02160 is highly analogous to the ALDH2 of brewer’s yeast (YMR170c) and is mainly responsible for ethanol oxidation and β-alanine biosynthesis. The FGSG_00979 protein contains a conserved domain in ALDH_F1-2_Ald2-like, which belongs to ALDH1 and ALDH2 and includes 10-formyltetrahydrofolate dehydrogenase, NAD+-dependent retinal dehydrogenase 1, and related proteins. The dissimilarity in the domain structure of the ALDH genes in *F. graminearum* could be the cause of their functional divergence.

### 3.2. ALDH Genes Have Various Roles in the Regulation of Acetaldehyde Dehydrogenase Activity

The ALDH family contains many homologous genes, which may have different enzyme activities due to their different structures. To examine the contribution of the ALDH genes to acetaldehyde dehydrogenase activity, we created deletion mutants, Δ*Fg04194*, Δ*Fg00139*, Δ*Fg11482*, Δ*Fg05375*, Δ*Fg00979*, Δ*Fg02160*, and Δ*Fg05831*, using a homologous recombination technique that involves the exchange of the ORF of the ALDH gene with a hygromycin B phosphotransferase gene cassette. The in vivo activity of acetaldehyde dehydrogenase in all seven ALDH-deletion mutants was monitored. The acetaldehyde dehydrogenase activity of Δ*Fg04194* was markedly increased, whereas that of Δ*Fg00979* was notably decreased. However, there were no considerable changes in the acetaldehyde dehydrogenase activity in the other six mutants (Figure 2). *FGSG_04194* and *FGSG_00979*, therefore, play major roles in regulating acetaldehyde dehydrogenase activity in *F. graminearum*.

### 3.3. ALDH Has an Effect on Mycelia Growth

To investigate the influence of ALDH on hyphal growth in *F. graminearum*, the mycelia growth of seven mutants were monitored on CM media. Except for mutants Δ*Fg00139* and Δ*Fg11482*, the other mutants grew significantly slower than the wild type in CM media (Figure 3A,B). In three days, the colony of the wild strain reached 58.50 mm in diameter, which was significantly larger than the colonies of the mutants, which ranged from 52.83 to 56.67 mm. Results demonstrate that ALDH genes have distinctive effects on the hyphal growth of *F. graminearum*.

### 3.4. ALDH Has a Role in Regulating Conidiogenesis in F. graminearum

Compared to those of PH-1, the conidia morphology of the ALDH-deletion mutants remained unchanged (Figure 4A). A significant decrease in conidia production by all mutants compared with the wild strain, which produced 3.28 × 10^6^ cfu/mL, was revealed. Among them, Δ*Fg05375* produced the most significant results, with a 51.1% reduction in conidia production, followed by Δ*Fg00139*, Δ*Fg04194*, Δ*Fg05831*, Δ*Fg00979*, Δ*Fg02160*, and Δ*Fg11482*, which exhibited 41.2%, 40.5%, 36.6%, 32.8%, 29.0%, and 26.7% decreases in conidia production, respectively. These results indicate that ALDH affects conidial production in *F. graminearum* (Figure 4B).

The germination of conidia from the ALDH knockout mutants was then analyzed. After 2 h, the PH-1 and ALDH knockout mutants began to germinate. In comparison to that of PH-1, the conidial germination of Δ*Fg04194* and Δ*Fg05375* was significantly delayed, whereas the other mutants did not show any significant differences in germination (Figure 4C,D). *FGSG_04194* and *FGSG_05375*, therefore, have a major impact on the conidial germination of *F. graminearum*.

### 3.5. ALDHs Regulate Environmental Stress Responses

Stress response assays indicated that at least one ALDH gene was involved in the response to each stressor. Δ*Fg04194* exhibited significantly increased sensitivity to 0.02% citral and 0.1% acetaldehyde but significantly decreased sensitivity to 0.01% SDS and 800 μg/mL CR. Compared to the wild type, Δ*Fg02160* displayed significantly decreased sensitivity to most of the stressors, apart from sorbitol. All the ALDH-deletion mutants, except Δ*Fg00139*, exhibited sensitivity to citral. Only 3% sorbitol promoted the growth of all the strains, whereas the other stressors suppressed their growth (Figure 5A,B). These results indicate that ALDH is critical for the response of *F. graminearum* to environmental stress.

### 3.6. ALDH Is Necessary for F. graminearum Pathogenicity and Deoxynivalenol Biosynthesis

Figure 6A demonstrates that all the ALDH-deletion mutants had a substantially lower pathogenicity in the wheat heads compared to the control, suggesting that the mutants were less aggressive.

DON accumulation in the five knockout mutant strains was significantly reduced after 21 days of incubation in wheat media. The toxin production of the mutant strains Δ*Fg04194*, Δ*Fg05375*, Δ*Fg00979*, and Δ*Fg05831* was significantly lower than that of the wild strain (2605.31 μg/kg) (Figure 6B), with decreases of 67.3%, 45%, 44.9%, and 31.3%, respectively. The Δ*Fg04194* strain had the lowest DON content. The findings suggest that ALDH plays a role in the DON biosynthesis of *F. graminearum*.

### 3.7. ALDH Is Important for Mitochondrial Function

To clarify the function of ALDH in the mitochondrial energy metabolism of *F. graminearum*, we examined the SDH activity of ALDH knockout mutants. Compared to that of PH-1, the SDH activity of the Δ*Fg05375* and Δ*Fg00979* mutants was similar; however, that of the Δ*Fg04194* mutant was significantly elevated. In contrast, the SDH activity of Δ*Fg00139*, Δ*Fg11482*, Δ*Fg02160*, and Δ*Fg05831* was significantly decreased (Figure 7A). Various ALDH genes, therefore, exert distinct regulatory effects on SDH.

The quantification data revealed that ATP production in Δ*Fg04194*, Δ*Fg05375*, Δ*Fg02160*, and Δ*Fg05831* was markedly higher than the control, and ATP content remained unchanged in the other mutants (Δ*Fg00139*, Δ*Fg11482*, and Δ*Fg00979*) (Figure 7B). These results indicate that the deletion of ALDH genes in *F. graminearum* affects the production of ATP. ALDH activity plays a critical role in maintaining critical mitochondrial function.

### 3.8. Transcriptomic Analysis of ΔFg04194

The transcriptome analysis identified 329 DEGs in the Δ*Fg04194* mutant compared to PH-1, with 263 genes being upregulated and 66 being downregulated. These results suggest that *FGSG_04194* affects the expression of multiple genes in *F. graminearum*. The volcano plot demonstrates that the DEGs had a symmetrical distribution around the mean. The abundance of the upregulated genes was greater than that of the downregulated genes (Figure 8A).

GO terms and KEGG pathways were used to determine the roles of the various DEGs. The top 20 GO terms of the DEGs in *F. graminearum* Δ*Fg04194* are shown in Figure 8B. The most highly enriched genes in molecular function (MF) were sphingosine hydroxylase, transmembrane transporter, and carbohydrate kinase activities. The cellular components (CC) in which the DEGs were mainly involved were the plasma membrane components. The biological processes (BP) in which the DEGs were mainly involved included carbohydrate metabolism, the regulation of inositol biosynthetic processes, and cell growth. These results indicate that most of the DEGs were associated with glucose and lipid metabolism.

The KEGG enrichment analysis further revealed the top 20 pathways associated with the differentially expressed genes in Δ*Fg04194*, which included starch and sucrose metabolism, glycolysis/gluconeogenesis, taurine metabolism, and hypotaurine metabolism (Figure 8C). We then tested the relative expression levels of seven typical DEGs using real-time qPCR, and the results were positively correlated with the transcriptomic data, confirming the credibility of the RNA-seq data (Figure 8D). To gain deeper insights into the mechanisms by which *FGSG_04194* regulates DON biosynthesis and pathogenicity, the DEGs associated with *FGSG_04194*-mediated pathogenicity and DON biosynthesis were identified (Table 1).

### 3.9. FGSG_04194 Is Localized to Lipid Droplets and Is Crucial for Their Accumulation

To determine the role of *FGSG_04194* in lipid metabolism, a strain carrying *FGSG_04194* tagged with GFP was constructed. The FGSG_04194-GFP strain was then utilized to confirm the subcellular localization of FGSG_04194 protein. As anticipated, strong GFP signals were observed in the mycelia and conidia of the FGSG_04194-GFP strain (Figure 9A). An experiment was conducted to investigate the co-localization of FGSG_04194-GFP and the LD indicator Nile Red, and the results indicated that the GFP and Nile Red signals were both present in the mycelia and conidia (Figure 9A), suggesting that FGSG_04194 is localized to LDs. To verify the changes in the LD content in the Δ*Fg04194* mutant, the mycelia and conidia of the PH-1, Δ*Fg04194*, and Δ*Fg04194-C* strains were stained with Nile Red and observed by a laser confocal microscope. Figure 9B,C demonstrate that the fluorescent signals in the conidia and hyphae of the Δ*Fg04194* strain were significantly weaker than those in the PH-1 and Δ*Fg04194-C* strains, suggesting that the deletion of *FGSG_04194* reduces LD accumulation.

## 4. Discussion

ALDH performs significant functions in various organisms, including bacteria, plants, mammals, and fungi [4,22,31,32]. ALDH is also involved in various stress responses [9]. Mycotoxin biosynthesis can be one of the functional responses of fungus to stress [33,34,35,36]. However, the effects and mechanisms of action of ALDH on secondary fungal metabolism, including mycotoxin production, remain unclear. Therefore, in this research, we assessed the possible function of ALDH in *F. graminearum*, which is a well-known pathogen of cereals and producer of DON.

ALDH has been identified as a crucial factor in vegetative growth, conidiogenesis, pathogenesis, and membrane stress tolerance in *M. oryzae.* The silencing of MoKDCDH and MoP5CDH in *M. oryzae* caused variations in fungal growth [21]. We identified seven ALDH genes and conducted related biological and functional analyses in *F. graminearum*. Several *ALDH*-deletion mutants were found to affect mycelial growth, suggesting that ALDH is important for the growth of nutrient-rich hyphae. ALDH is also critical for the formation of conidia, stress responses, and pathogenicity of *F. graminearum*, which is consistent with the established role of ALDH in *M. oryzae*. ALDH significantly influences the production of DON. Our results indicated that ALDH can influence toxin production by fungi. Our findings, therefore, further elucidate the function of ALDH and can be used for studying the regulation of DON toxins. The specific regulation and targeting of ALDH biosynthetic pathways may be a possible solution for treating plant and animal diseases caused by certain phytopathogenic fungi, as suggested by Singh et al. [9].

Through oxidation and dehydrogenation, ALDH can convert aldehydes into nontoxic carboxylic acids, which provide protection and detoxification to living organisms but are also associated with stress. ALDHs are upregulated in response to various forms of stress in organisms, including bacteria (due to environmental and chemical stress) [32,37], plants (such as dehydration, salinity, and oxidative stress) [15,38,39], yeast (ethanol exposure and oxidative stress) [18,40], mammals (from oxidative stress and lipid peroxidation) [41], and *M. oryzae* (which promotes oxidative and reductive stress tolerance) [21]. In this study, we examined the responses of seven ALDH-deficient mutants to different stressors. The different ALDH-deficient mutants responded inconsistently to the stressors, and not every ALDH was found to play a role in each form of stress. In culture media supplemented with acetaldehyde, only Δ*Fg04194*, Δ*Fg11482,* and Δ*Fg02160* displayed notable sensitivity to the compound, whereas the other mutants did not show any considerable alterations. This implies that not all ALDH genes are involved in acetaldehyde transformation. All seven gene deletion mutants, except Δ*Fg00139*, exhibited heightened sensitivity to citral in the media, suggesting that most ALDH genes are implicated in the reaction to citral stress. Research investigating the response of cells to stress has highlighted the necessity and individual significance of each ALDH gene in preserving internal balance. Moreover, citral can be used as an inhibitor of ALDH, which indicates a possible strategy for controlling the propagation of *F. graminearum*.

DON is regarded as the most essential factor in the pathogenicity of *F. graminearum* [24]. Our research showed that the pathogenicity of *F. graminearum* was significantly reduced in all seven ALDH-deletion mutants, and the production of DON was only decreased in four ALDH-deletion mutants, which emphasizes the importance of ALDH in both pathogenicity and secondary metabolite production. The lower aggressiveness of these ALDH-deficient mutants may be linked to the slower mycelial growth and fewer spores observed in this study, as well as to a reduction in DON production as reported in the literature [42]. Previously, the considerable influence of ALDH on biosynthetic pathways and energy production was demonstrated [9]. We formerly thought that the absence of the ALDH gene caused a reduction in acetaldehyde dehydrogenase activity, thus resulting in a decrease in toxin production. However, this study has revealed that the activity of acetaldehyde dehydrogenase and succinate dehydrogenase in the Δ*Fg04194* mutant increased, along with an increase in the ATP content. The transcriptome sequencing results confirm the observation that energy-metabolism-related genes are upregulated in the Δ*Fg04194* mutant. Research has revealed that the absence of *FGSG_04194* does not reduce acetaldehyde dehydrogenase activity; however, its absence can increase the gene expression and activity of other acetaldehyde dehydrogenases. Many plant ALDHs are expressed constitutively, while others are activated in response to various environmental and biological stressors, as has been reported in the literature [9]. The DON toxin level observed in the Δ*Fg00979* mutant was lower, and its acetaldehyde dehydrogenase enzyme activity was also decreased, indicating the necessity of *FGSG_00979* for acetaldehyde dehydrogenase enzyme activity. When the acetaldehyde dehydrogenase activity was deficient, the production of acetyl-CoA and synthesis of toxins decreased, which could be the mechanisms underlying the reduction in DON production and pathogenicity in the Δ*Fg00979* mutant. The single-gene-deletion Δ*Fg05375* and Δ*Fg05831* mutant strains showed a decrease in DON synthesis, yet their acetaldehyde dehydrogenase enzyme activity remained unchanged, suggesting that these two ALDH genes had no influence on acetaldehyde dehydrogenase activity. Therefore, the relationship between acetaldehyde dehydrogenase activity and toxin synthesis can be complex and context-dependent. It involves understanding the specific ALDH isoforms expressed, their substrate specificity, and the regulatory factors that control their activity under different conditions. It seems to be that certain ALDH genes will be activated under special conditions. Further research is needed to elucidate the precise mechanisms and relationships between ALDH enzymes and toxin synthesis.

The generation of fungal secondary metabolites is likely influenced by the energy status of mitochondria, which primarily produce ATP through oxidative phosphorylation [43]. Acetaldehyde dehydrogenase is a critical enzyme involved in alcohol metabolism and mitochondrial oxidative ATP production [44]. SDH plays a key role in the mitochondrial electron transport chain and tricarboxylic acid cycle [45]. This study was conducted to measure the activities of acetaldehyde dehydrogenase, SDH, and ATP in ALDH-deletion mutants. Our results showed a stark contrast between the acetaldehyde dehydrogenase activity of Δ*Fg04194* and Δ*Fg00979*, with there being a significant decrease in the latter and significant increase in the former, implying that *FGSG_00979* is essential for acetaldehyde dehydrogenase activity, whereas *FGSG_04194* has an inhibitory effect on acetaldehyde dehydrogenase activity. The other ALDH-deletion mutants did not affect acetaldehyde dehydrogenase activity, suggesting that they may be involved in other biological processes. Acetaldehyde dehydrogenase activity, SDH activity, and the ATP content increased, and the expression of many energy-related genes was upregulated in Δ*Fg04194*, suggesting that *FGSG_04194* plays a significant role in the inhibition of mitochondrial energy-metabolism-associated pathways. In Δ*Fg02160 and* Δ*Fg05831*, the SDH level decreased while the ATP content increased, demonstrating that the acetaldehyde dehydrogenase activity, SDH activity, and ATP contents were all affected by the ALDH genes. These data suggest that ALDH is a key factor in mitochondrial function and is involved in energy metabolism in *F. graminearum*.

LDs are dynamic organelles that are essential for the regulation of lipid metabolism [46]. LDs are essential for the virulence of pathogenic fungi. In *Candida parapsilosis* and *Metarhizium acridum*, the growth and maturation of pathogens are hindered when the synthesis of LDs is disrupted, which causes a significant reduction in their pathogenicity [47,48]. Disruptions in the synthesis or degradation of LDs in *Magnaporthe oryzae* can have a detrimental effect on appressorium formation, thus significantly reducing the capacity of the fungus to cause diseases in rice [49]. Our study revealed that FGSG_04194 protein is localized in LDs and is only visible in the hyphae or germinating spores. Δ*Fg04194* led to a decrease in LD synthesis, implying its involvement in lipid metabolism. As a result of the gene deletion, the amount of toxin present was reduced, and the pathogenicity was weakened, which could have been attributed to the disruption of LD synthesis.

*FGSG_04194* is involved in various stress responses, with there being a marked decrease in DON toxins and a notable increase in ALDH and SDH enzyme activities and ATP content upon deletion, indicating its multiple biological functions. Transcriptome sequencing revealed that more genes were upregulated than downregulated, suggesting that FGSG_04194 likely inhibits various metabolic pathways.

Transcription factors (TFs) that regulate pigmentation [50], mycotoxin biosynthesis [51,52,53], sexual development, and virulence have been characterized in *F. graminearum* [54,55]. In this study, the expression of three TF genes (*FGSG_00217*, *FGSG_03873*, and *FGSG_03649*) was downregulated and that of *FGSG_00713* was upregulated in Δ*Fg04194*. One previous study demonstrated that Δ*Fg00217* and Δ*Fg03873* showed a colony morphology similar to that of the wild type. However, *FGSG_00217* was upregulated and *FGSG_03873* was downregulated following FgV1 infection [56,57]. The FgV1 infection affected the expression levels of *FGSG_00217* and *FGSG_03873*. *FGSG_03649* was downregulated 96 h after infection in the susceptible wheat cultivar, suggesting that *FGSG_03649* is involved in pathogenic invasion [58]. The weakened pathogenicity of Δ*Fg04194* in this study may have been due to the involvement of three TFs in regulating the infective ability of *F. graminearum*. *FGSG_00713* is a stress-related TF [59]. In this study, the upregulation of *FGSG_00713* in the Δ*Fg04194* mutant may indicate that this gene is involved in stress response tolerance.

In fungi that cause plant diseases, transporters are particularly responsible for transporting endogenous toxins, plant-defense compounds, and fungicides. The two major classes of transporters involved in the secretion of these compounds are the ATP-binding cassette (ABC) and major facilitator superfamily (MFS) transporters [60]. ABC transporters are responsible for fungal fitness and virulence. ABC-C group V of *Aspergillus* is responsible for transporting fungal secondary metabolites [61]. Deletion mutants of FgABC1 in *F. graminearum* are virulent in wheat, barley, and maize, suggesting that FgABC1 is responsible for secreting secondary fungal metabolites [62]. In the present study, *FGSG_02263*, which encodes an ABC-type transporter, was found to be downregulated in Δ*Fg04194*. The encoding MFS-type efflux pump, *FGSG_03571*, was significantly downregulated, as were *FGSG_09595* and *FGSG_10923*. *FGSG_09595* encodes toxin efflux pump genes in *F. graminearum* [63]. The toxin content of the Δ*Fg04194* mutant may have decreased, possibly associated with ABC and MFS transporters. The decreased expression of ABC and MFS transporters can impede the transportation of toxins, thereby reducing their pathogenicity. *FGSG_03882* is an ABC transporter that contains nucleotide-binding domains and a transmembrane-spanning domain that encodes pleiotropic drug resistance proteins. *FGSG_03882* expression has been found to be downregulated after tebuconazole treatment [64]. In the present study, *FGSG_03882* expression was significantly upregulated in the *FGSG_04194* deletion mutant. Furthermore, the expression of *FGSG_02966* and *FGSG_07802*, which encode efflux pumps, was upregulated. The upregulation of these three transporters in Δ*Fg04194* may have been correlated with the response of the mutant to multiple stressors. This suggests that transporters are involved in the transport and expulsion of substances from the cell. *FGSG_04194* regulates the expression of drug targets and efflux pump transporters, which are the main contributors to multiple chemical resistance.

The cell walls of fungi serve as the external layers of the fungal cells, shielding them from unfavorable environmental conditions. Chitin is an essential element in the cell walls of most filamentous fungi, providing structural support [65]. The nine chitin synthetase genes have different functions in *F. graminearum*. GzCHS5 and GzCHS7 play important roles in fungal growth, development, and pathogenicity [66]. FgCHS8 affects virulence and fungal cell wall sensitivity to environmental stress [67]. In this study, the expression of the chitin synthase gene *FGSG_03418* and the chitin synthase regulatory gene *FGSG_08673* was significantly upregulated in Δ*Fg04194*, whereas that of chitin deacetylase *FGSG_03544* was downregulated. This may have led to enhanced chitin synthesis, which may have increased cell wall resistance and tolerance, allowing *F. graminearum* to adapt to various environmental stressors. This was consistent with the tolerance of Δ*Fg04194* to SDS and CR stress. The *FGSG_04194* knockout mutant showed significant tolerance to cell wall stressors, indicating a non-redundant role for ALDH in the fungal adaptation to stress.

## 5. Conclusions

In conclusion, we identified 29 proteins that are homologous to the yeast aldehyde dehydrogenase ALD2 from *F. graminearum*, nine of which were classified as genes that code for aldehyde dehydrogenase. By employing homologous recombination, seven ALDH genes were successfully knocked out and their roles were characterized. The deletion of these seven genes resulted in a decrease in conidia production and a weakened pathogenicity. Two ALDH genes, *FGSG_04194* and *FGSG_00979*, were observed to be associated with acetaldehyde dehydrogenase activity, while five ALDH genes, namely *FGSG_04194*, *FGSG_05375*, *FGSG_00979*, *FGSG_02160*, and *FGSG_05831*, were found to be related to mycelial growth. All gene deletion mutants showed varying reactions to different stresses. Four ALDH genes (*FGSG_04194*, *FGSG_05375*, *FGSG_00979*, and *FGSG_05831*) were found to have an impact on DON production, while five ALDH genes (*FGSG_04194*, *FGSG_00139*, *FGSG_11482*, *FGSG_02160*, and *FGSG_05831*) were associated with SDH activity, and four ALDH genes (*FGSG_04194*, *FGSG_05375*, *FGSG_02160*, and *FGSG_05831*) were linked to ATP production. We also provide evidence that *FGSG_04194* may be involved in lipid metabolism. However, further research is required to determine the subcellular localization of other ALDHs and the intrinsic molecular mechanisms associated with pathogenicity and mycotoxin synthesis in fungi. We believe that ALDH is an important target for the control of fungal diseases in cereals.

## Figures and Tables

**Figure 1 microorganisms-11-02875-f001:**
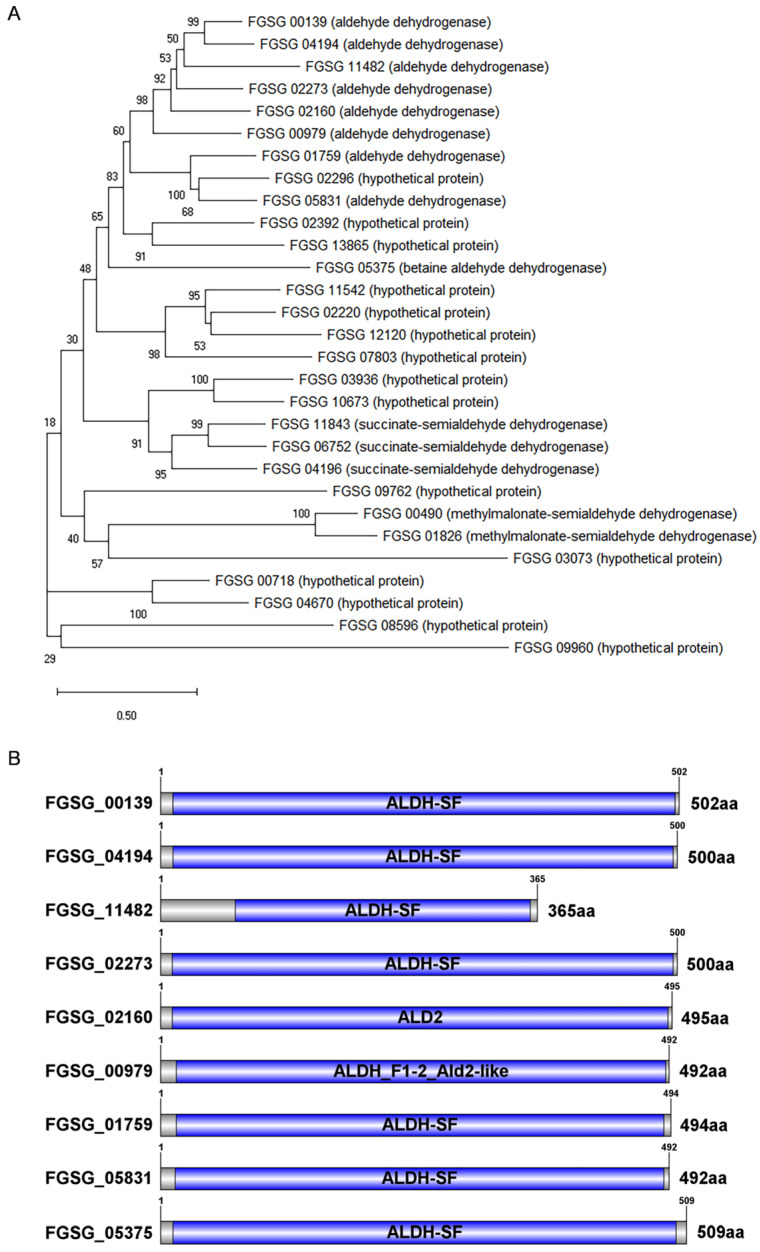
Phylogenetic analysis and conserved domain prediction of ALDHs in *F. graminearum*. (**A**) Phylogenetic analysis of 29 ALD2 homologous proteins from *F. graminearum*. CLUSTALW was employed to align the full-length amino acid sequences of FgALDH. The phylogenetic tree was created by utilizing the neighbor-joining method through MEGA-X software. The numerical values assigned to the nodes in the rooted tree are the bootstrap values determined through 1000 replications. The bar illustrates a distance of 0.50 units. (**B**) Prediction of conserved domains for nine aldehyde dehydrogenases from *F. graminearum*. NCBI CDD was utilized to search for conserved domains using the ALDH protein sequence. The conserved domains were marked with a blue box.

**Figure 2 microorganisms-11-02875-f002:**
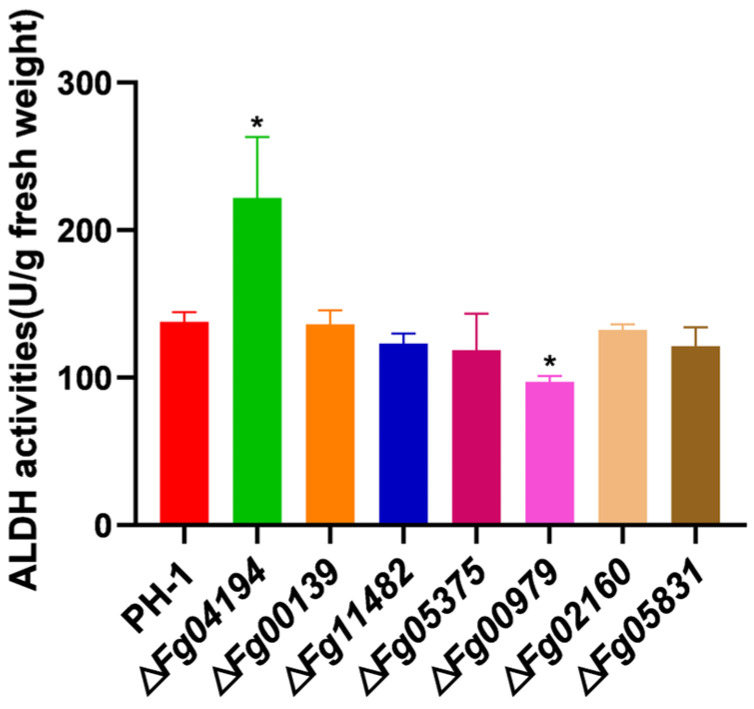
Effect of ALDH on acetaldehyde dehydrogenase activity in *F. graminearum*. Acetaldehyde dehydrogenase activity of wild-type PH-1 and ALDH-deletion mutants grown on CM for 72 h. Significance marked using “*” (*p* < 0.05).

**Figure 3 microorganisms-11-02875-f003:**
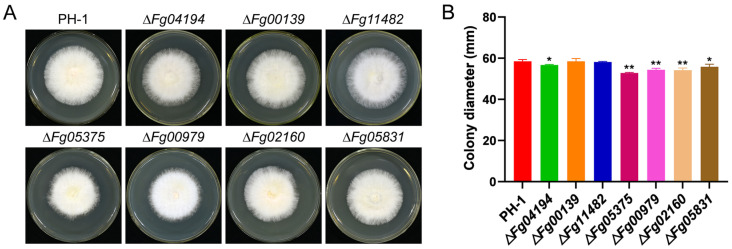
ALDH has an effect on the growth of *F. graminearum*. (**A**) Colony morphology of wild-type and ALDH knockout mutant strains cultured for 72 h on CM medium. (**B**) Statistical analysis of growth defects exhibited by seven independent ALDH knockout mutants compared to the wild-type strain. Significance marked using “*” (*p* < 0.05) and “**” (*p* < 0.01).

**Figure 4 microorganisms-11-02875-f004:**
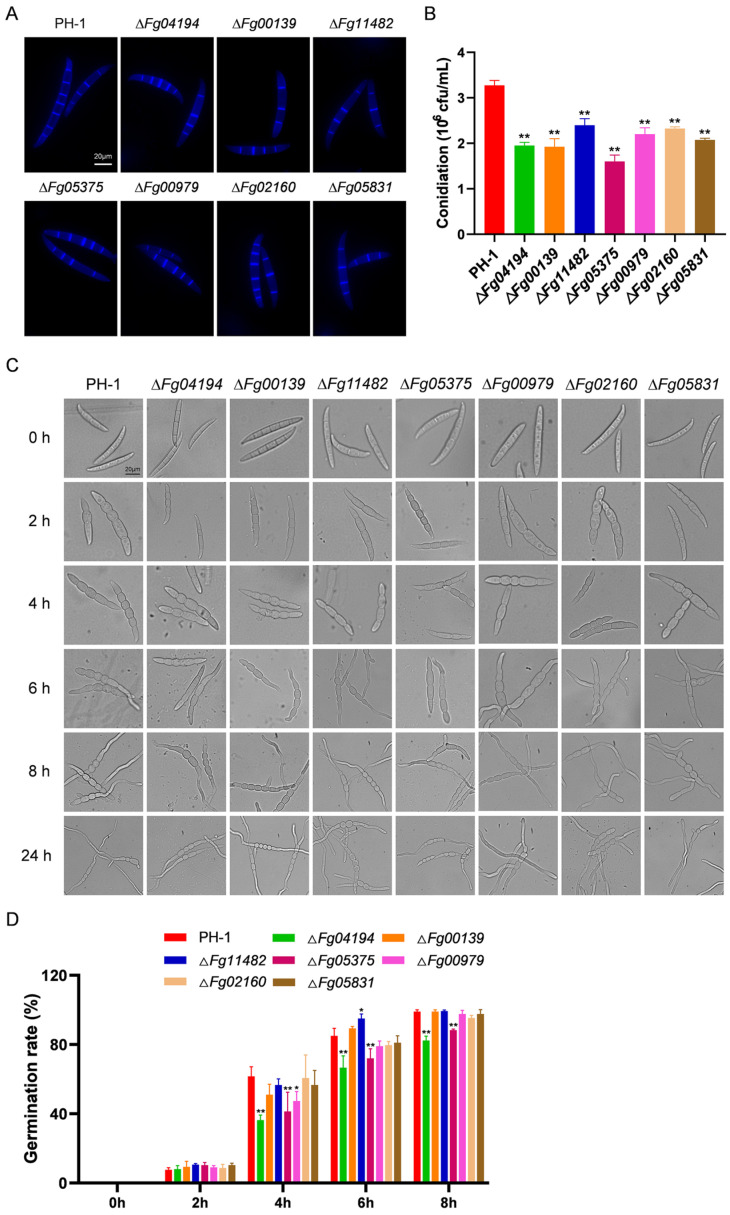
ALDHs play a significant role in the conidiogenesis of *F. graminearum*. (**A**) Conidia morphology of wild-type PH-1 and ALDH-deletion mutants grown on CMC for 72 h were stained with CFW. (**B**) Conidia production of PH-1 and ALDH-deletion mutants. (**C**) Germination of conidia from seven independent ALDH-deletion mutants at different time periods. (**D**) Germination rate of each ALDH-deletion mutant in each time period. Significance marked using “*” (*p* < 0.05) and “**” (*p* < 0.01).

**Figure 5 microorganisms-11-02875-f005:**
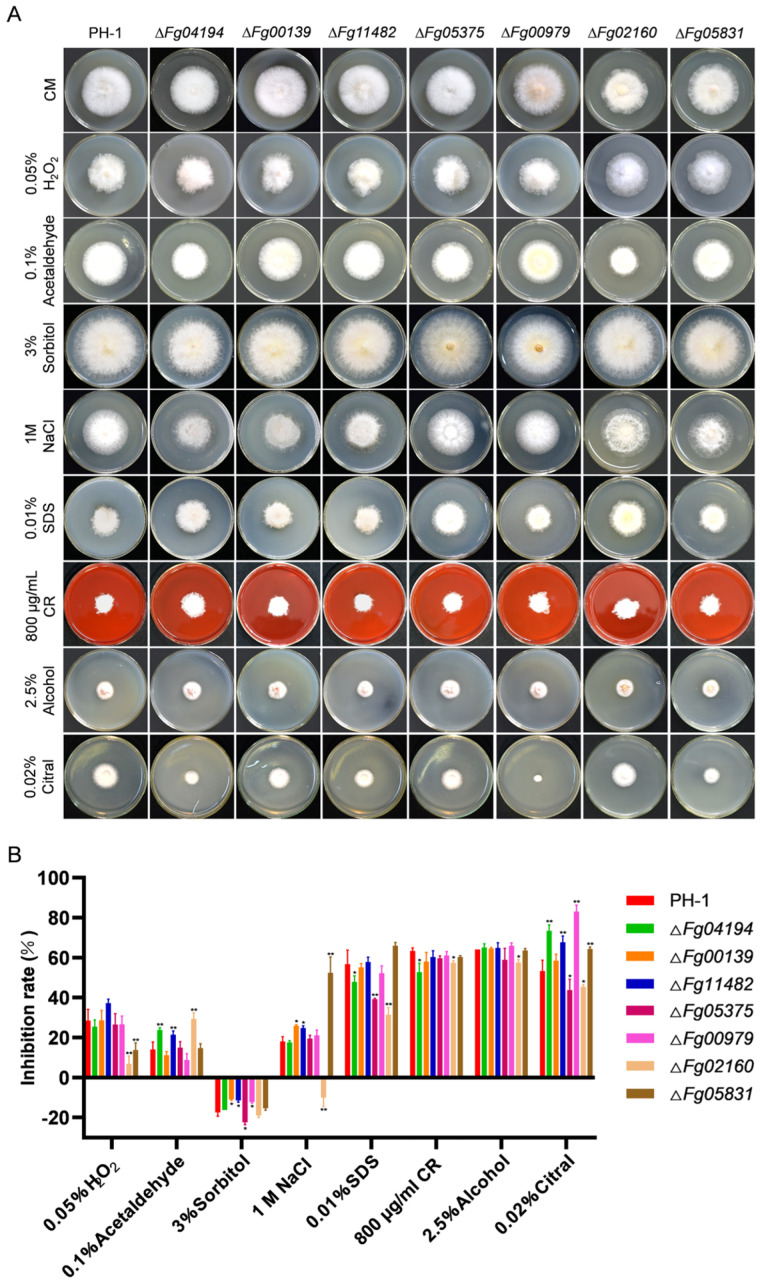
ALDHs contribute to stress responses in *F. graminearum*. (**A**) Growth of PH-1 and ALDH-deletion mutants on CM containing 0.05% H_2_O_2_, 0.1% acetaldehyde, 3% sorbitol, 1 M NaCl, 0.01% sodium dodecyl sulfate (SDS), 800 μg/mL Congo Red (CR), 2.5% alcohol, and 0.02% citral. (**B**) The inhibition rate of the PH-1 and ALDH-deletion mutants on CM supplemented with 0.05% H_2_O_2_, 0.1% acetaldehyde, 3% sorbitol, 1 M NaCl, 0.01% SDS, 800 μg/mL CR, 2.5% alcohol, and 0.02% citral. Inhibition rate = [(diameter of unstressed strain − diameter of stressed strain)/(diameter of unstressed strain)] × 100%. Significance marked using “*” (*p* < 0.05) and “**” (*p* < 0.01).

**Figure 6 microorganisms-11-02875-f006:**
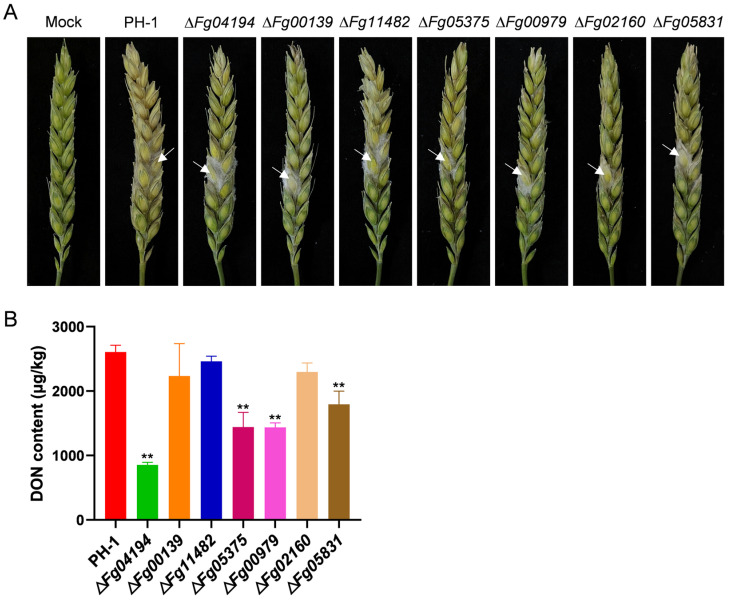
Effects of ALDH on pathogenicity and DON production. (**A**) Infectivity of mutants on wheat spikes for 10 days compared with that of PH-1. Arrows mark the location of injections of wheat ears. (**B**) DON content of wild-type and ALDH-deletion mutants cultured in wheat media for 21 days. Significance marked using “**” (*p* < 0.01).

**Figure 7 microorganisms-11-02875-f007:**
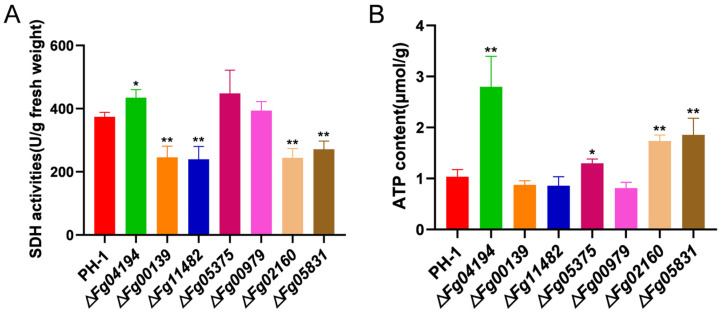
(**A**) SDH activity of PH-1 and ALDH-deletion mutants grown on CM for 72 h. (**B**) ATP content of PH-1 and ALDH-deletion mutants grown on CM for 72 h. Significance marked using “*” (*p* < 0.05) and “**” (*p* < 0.01).

**Figure 8 microorganisms-11-02875-f008:**
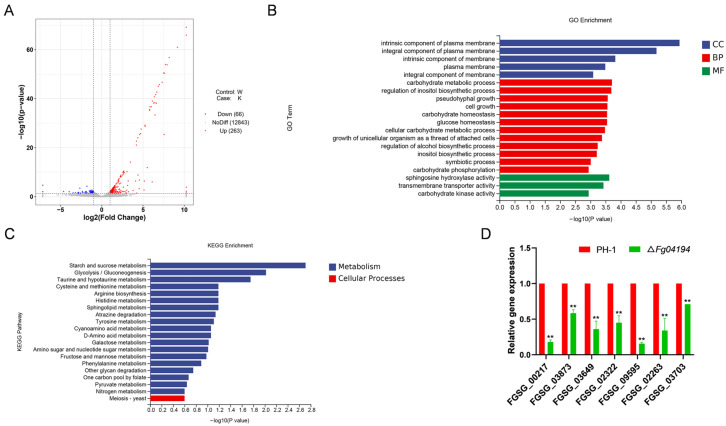
Transcriptomic analysis of PH-1 compared to Δ*Fg04194*. (**A**) Volcano plots of DEGs. (**B**) GO enrichment analysis of DEGs (*Padj* < 0.05). (**C**) KEGG enrichment analysis of DEGs (*Padj* < 0.05). (**D**) Relative expression levels of transcription factors and toxin-related genes in wild-type PH-1 and the Δ*Fg04194* knockout mutant cultured in CM for 72 h. The internal reference gene is tubulin. Significance marked using “**” (*p* < 0.01).

**Figure 9 microorganisms-11-02875-f009:**
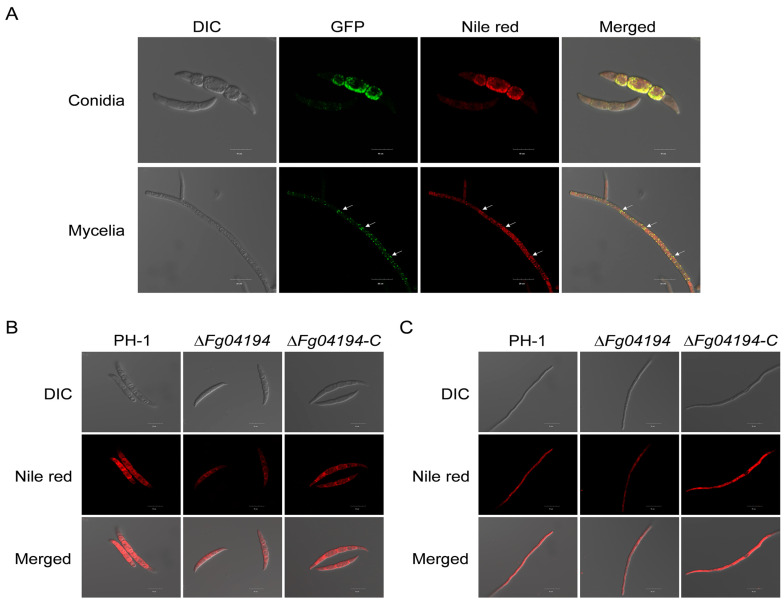
Subcellular localization of the FGSG_04194 protein and the accumulation of lipid droplets in Δ*Fg04194*. (**A**) Nile Red staining of conidia and mycelia of the FGSG_04194-GFP complementation (Δ*Fg04194-C*) strain. Arrows indicate the lipid droplets localization of FGSG_04194-GFP. Bar = 10 µm for conidia panels, bar = 20 µm for mycelia panels. (**B**) Nile Red staining of the conidia of strains PH-1, Δ*Fg04194*, and Δ*Fg04194-C*. Bar = 10 µm. (**C**) Nile Red staining of mycelia from strains PH-1, Δ*Fg04194*, and Δ*Fg04194-C*. Bar = 15 µm.

**Table 1 microorganisms-11-02875-t001:** RNA-seq analysis of DEGs in the Δ*Fg04194* and wild-type strains.

Gene Category	Log_2_(fc)	*p*-Value	Function
Transcription factors			
*FGSG_00217*	−1.0766	3.25 × 10^−2^	Transcription factors
*FGSG_03873*	−1.1957	4.47 × 10^−2^	Transcription factors
*FGSG_03649*	−1.3199	4.25 × 10^−2^	Transcription factors
*FGSG_00713*	1.1959	5.61 × 10^−3^	Transcription factors
Aurofusarin biosynthesis gene cluster			
*FGSG_02322*	−1.2654	3.06 × 10^−2^	GIP4/AurT
*FGSG_02323*	−1.3355	1.17 × 10^−2^	GIP5/AurR2
Transporter-related genes			
*FGSG_02263*	−2.8354	2.78 × 10^−2^	ABC-type transporter
*FGSG_03571*	−3.4838	2.88 × 10^−2^	MFS-type efflux pump MFS1
*FGSG_09595*	−1.9665	3.41 × 10^−2^	MFS-type transporter
*FGSG_10923*	−1.7986	5.82 × 10^−5^	Efflux pump himE
*FGSG_03882*	1.1826	1.03 × 10^−2^	ABC multidrug transporter
*FGSG_02966*	1.4650	2.21 × 10^−4^	MFS-type efflux pump MFS2
*FGSG_07802*	1.1539	3.86 × 10^−4^	Efflux pump FUS6
Hydrophobic protein			
*FGSG_09066*	1.0376	2.92 × 10^−2^	Hydrophobin 3 precursor
Chitin-synthesis-related genes			
*FGSG_03418*	1.4372	1.48 × 10^−4^	Chitin synthase 1
*FGSG_08673*	1.4138	2.16 × 10^−4^	Chitin synthase regulator 2
*FGSG_03544*	−1.0391	5.29 × 10^−3^	Chitin deacetylase
Energy-metabolism-related genes			
*FGSG_08774*	1.4856	3.92 × 10^−4^	Glucokinase
*FGSG_08399*	1.2927	4.51 × 10^−3^	Hexokinase-1
*FGSG_01743*	1.4074	9.58 × 10^−4^	Acetyl-coenzyme A synthetase
*FGSG_08343*	1.2453	5.91 × 10^−3^	Plasma membrane ATPase
*FGSG_01522*	1.9596	2.29 × 10^−6^	Adenylate cyclase
Other protein-related genes			
*FGSG_01582*	6.1298	1.01 × 10^−39^	RNA-dependent RNA polymerase 1
*FGSG_04619*	6.4425	4.17 × 10^−39^	RNA-dependent RNA polymerase 1
*FGSG_13459*	−4.6314	8.34 × 10^−3^	Cytochrome P450 monooxygenase
*FGSG_03264*	−1.9838	1.02 × 10^−2^	Cytochrome P450 monooxygenase

## Data Availability

PacBio and Illumina Sequencing data were submitted to the National Center for Biotechnology Information (NCBI) Sequence Read Archive (SRA) under Bio-Project: PRJNA983120.

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
