# Peer review of "Functional Characterization of Aldehyde Dehydrogenase in *Fusarium graminearum"

_microorganisms, 2023, doi:10.3390/microorganisms11122875_

Round 1
Reviewer 1 Report
Comments and Suggestions for Authors
Dear Authors
The present article entitled “Functional characterization of aldehyde dehydrogenase in Fusarium graminearum” is an interesting article for the biological functions of the ALDHs indicated that their role in the conidial production, DON regulation, stress responses, and pathogenicity of F. graminearum. The ALDHs as an oxidoreductase group insert bonds into extracellular proteins. These proteins could be important virulence factors. Therefore, it is interesting to investigate their possible pathogenicity and other functions.
The idea of the experiment is respectable and the experiments seem organized.
Figures are of good quality. Figure 1 needs a better, more complete legend.
The conclusion could be better; I mean to be sharper and richer, taking into account the large amount of work and results that this study has to show.
Authors should state in the article which species and variety of wheat they have used, and whether it is susceptible to F. graminearum infection in general.
My overall Recommendation is to accept the article after minor revision

As far as I can tell, the Quality of the English language does not seem to have a particular problem.
Author Response
Dear Reviewer,
We really appreciated sincerely your careful review and positive comments on our manuscript (microorganisms-2682965) “Functional characterization of aldehyde dehydrogenase in Fusarium graminearum”. Thank you for giving us the precious opportunity to revise our manuscript. According to the comments, we checked the manuscript thoroughly and made relevant changes. Please find the comments in black, followed by our responses in red. Please see the attachment.

Reviewer 2 Report
Comments and Suggestions for Authors
The article is about the demonstration of the physiological role of aldehyde dehydrogenases of Fusarium graminearum. Nine possible ALDH genes were detected in the genome on the basis of homologies to the yeast Saccharomyces cerevisiae.
Basic problems:
I can not see why not a closer relative fungal genome was the basis of the search.
If we appreciated the nine genes and mutated them one by one, it would have been interesting to analyze the expression rate of the non-mutated genes. Based on the article, several physiological changes happen if one of the genes is mutated, and even the ALDH activities change up or down in the strains, showing us a parallel expression change in another ALDH gene(s).
Therefore, I can take 3.1, 3.2, 3.3, 3.4, 3.5, 3.6, 3.7, 3.8, and 3.9 as valid results, with some comments; however, I see severe trouble in Discussion (4). I would be very hesitant to discuss an enzyme-coding gene as a regulatory gene. Please take the biochemical role and the total ALDH activities into consideration. It seems to be that different genes will be activated within other conditions.
Further comments:
Lane 70 -74: This DON part is too forward; you do not need to express it.
Lane 157: how do you select four pieces of Mycelium?
Please clarify 2.7., something is missing on centrifugation.
Figure 2: I think the standard deviation is extremely low. How do you measure the colony size?
3.5. We need the wheat type. Also in the Methods. Was it a hybrid without endophyte? Was it a pot experiment?
3.6 I would put these parts after the genetic analyses. And it should be mentioned that we do not know which gene was expressed.
In 3.7, Lane 328. ALDH does not control ATP production. However, its activity is needed to maintain important biochemical functions.
Also at lane 354.
Figure 7A, we do not need this part.
Lane 441: Please do not mix aflatoxins to this topic.
Author Response

(The authors gave the same response as above.)
